# DIFFERENTIAL EQUATION NETWORKS

## ABSTRACT

Most deep neural networks use simple, fixed activation functions, such as sigmoids or rectified linear units, regardless of domain or network structure. We introduce differential equation networks, an improvement to modern neural networks in which each neuron learns the particular nonlinear activation function that it requires. We show that enabling each neuron with the ability to learn its own activation function results in a more compact network capable of achieving comperable, if not superior performance when compared to much larger networks. We also showcase the capability of a differential equation neuron to learn behaviors, such as oscillation, currently only obtainable by a large group of neurons. The ability of differential equation networks to essentially compress a large neural network, without loss of overall performance makes them suitable for on-device applications, where predictions must be computed locally. Our experimental evaluation of real-world and toy datasets show that differential equation networks outperform fixed activatoin networks in several areas.

## 1 INTRODUCTION AND RELATED WORK

Driven in large part by advancements in storage, processing, and parallel computing, deep neural networks (DNNs) have become capable of outperforming other methods across a wide range of highly complex tasks. Although DNNs often produce better results than shallow methods from a performance perspective, one of the main drawbacks of DNNs in practice is computational expense. One could attribute much of the success of deep learning in recent years to cloud computing, and GPU processing. While deep learning based applications continue to be integrated into all aspects of modern life, future advancements will continue to be dependent on the ability to perform more operations, faster, and in parallel unless we make fundamental changes to the way these systems learn.

State-of-the-art DNNs for computer vision, speech recognition, and healthcare applications require too much memory, computation, and power to be run on current mobile or wearable devices. To run such applications on mobile, or other resource-constrained devices, either we need to use these devices as terminals and rely on cloud resources to do the heavy lifting, or we have to find a way to make DNNs more compact. For example, ProjectionNet by Ravi (Ravi, 2017), and MobileNet by Howard et al. (Howard et al., 2017) are both examples of methods that use a compact DNN representations with the goal of on-device applications. In ProjectionNet, a compact "projection" network is trained in parallel to the primary network, and used for the on-device network tasks. MobileNet proposes a streamlined architecture in order to achieve network compactness. One drawback to these approaches is that network compactness is achieved at the expense of performance. In this paper, we propose a different method for learning compact, powerful, stand-alone networks; we allow each neuron to learn its individual activation function.

Up to this point, advancements in deep learning have been largely associated with one of four areas: optimization, regularization, activation, and network structure. Some of these advancements, such as network structure, address issues related to the type of problem at hand, and the nature of the data. Since the advent of the classic feed-forward neural network (FFNN), researchers have introduced highly effective network structures such as convolutional neural networks (CNN (Krizhevsky et al., 2012), popular in computer vision, and Recurrent Neural Networks (RNN) for sequences. The advent of new activation functions such as rectified linear units (ReLU) (Nair & Hinton, 2010), exponential linear units (ELU) (Clevert et al., 2015), and scaled exponential linear units (SELU) (Klambauer et al., 2017) address a networks ability to effectively learn complicated functions, thereby allowing

them to perform more complicated tasks. The choice of an activation function is typically determined empirically by tuning, or due to necessity. For example, in modern deep networks, the ReLU activation function is often favored over the sigmoid function, which used to be a popular choice in the earlier days of neural networks. A reason for this preference is that the ReLU function is non-saturating and does not have the vanishing gradient problem when used in deep structures (Hochreiter, 1998). Recently, searching algorithms have proven to be valuable for automatic construction of previously hand-designed network components and activation functions (Zoph et al., 2017; Bello et al., 2017; Ramachandran et al., 2017). Zoph et al. (2017); Bello et al. (2017); Ramachandran et al. (2017) have proposed one such method for searching for more effective activation functions.

At first glance, and from an applicability perspective, our approach is similar to max-out networks (Goodfellow et al., 2013), adaptive piecewise linear units by Agostinelli et al. (2014), and Ramachandran et al. (2017). However, in Ramachandran et al. (2017), their functions do not transform themselves during network training, and all neurons utilize the same activation function. We introduce differential equation networks (DifENs), where the activation function of each neuron is the nonlinear, possibly periodic solution of a second order, linear, ordinary differential equation. While the number of parameters learned by max-out and PLU is proportional to the number of input weights to a neuron, and the number of linear units in that neuron, for each DifEN activation function we learn only five additional parameters.

One feature of DifENs that is particularly interesting within the healthcare industry is the ability of DifENs to regress decaying and periodic functions. Although efforts have been made to explore the usefulness of periodic functions in neural networks since the 1990s, their applicability has not yet been appreciated (Sopena et al., 1999). Thanks to the successes of deep learning in recent years, researchers have begun re-exploring the potential of periodic functions as activations (Parascandolo et al., 2016). This capability makes DifENs particularly well suited to problems where the desired output signals display combinations of periodicity and decay. Some examples include disease management, modeling chronic conditions such as exacerbation of COPD, medication adherence, and acute conditions that are expected to get better over time such as cancer with a successful course of treatment. Inversely, healthiness indices that might decay over time, and that could drastically effect a patients health if effectively modelled (e.g. cancer with an unsuccessful course of treatment).

Our contributions in this paper include the following: We introduce differential equation networks, which can learn complex concepts in compact representations. We propose a learning process to learn the parameters of a differential equation for each neuron. Finally, we empirically show that a differential equation network is effective when applied to real-world problems, and that they are capable of changing activation functions during the learning process.

## 2 NETWORK COMPRESSION

The size of a neural network is delineated its number of hidden neurons and their interconnections, which together determine the network's complexity. The ideal size of a network depends on the intricacy of the concept it is required to learn. A network that is too small cannot entirely, and circumstantially learn a hard problem.

The existing, successful structures of neural networks Sutskever et al. (2014); LeCun et al. (2015); Bahdanau et al. (2014) use hundreds of thousands to millions of parameters to learn common tasks, such as translation or object recognition. Although these tasks are complex to model, and still challenging for modern neural networks (Sutskever et al., 2014), they are considered "easy" when performed by a qualified human. It follows that, if we want to create a neural network with abilities that are beyond average human level aptitude, the size of such a network would need to grow considerably. For example, we have seen accuracy improvements in state-of-the-art DNNs applied to massive image classification problems resulting in part from a large increase in the number of layers He et al. (2016). If this trend continues to subsist, shortly, we will witness neural networks with billions of parameters that are designed for sophisticated tasks such as live speech understanding and translation, as well as controlling robots or humanoids, which would require an immense increase in storage and computational resources.

For these reasons, we want to highlight the importance of efficient network design, and the opportunities presented by network contraction methodologies. We show that differential equation networks learn a given task with a smaller network when compared to a fixed activation DNNs.

## 3 METHOD

Inspired by functional analysis and calculus of variations, instead of using a fixed activation function for each layer, we propose a novel solution for learning an activation function for each neuron in the network. In the results section we show that by allowing each neuron to learn its own activation function, the network as a whole can perform on par with (or even outperform) much larger baseline networks. In the supplementary material, we have briefly outlined how in Calculus of Variations, the Euler-Lagrange formula can be used to derive a differential equation from an optimization problem with function variables (Gelfand & Fomin, 1963; Gelfand et al., 2000).

The idea of this paper is to find the parameters of an ordinary differential equation (ODE) for each neuron in the network, whose solution would be used as the activation function of the neuron. As a result, the differential equation network enables each neuron to learn a personalized activation function flexibly. We select (learn) the parameters of the differential equation from a low dimensional space (i.e., five). By minimizing the network loss function, our learning algorithm smoothly updates the parameters of the ODE, which would result in an uncountably [1] extensive range of possible activation functions.

### 3.1 DIFFERENTIAL EQUATION ACTIVATION

We parametrize the activation function of each neuron using a linear, second order ordinary differentiatial equation $ay''(t) + by'(t) + cy(t) = g(t)$, parameterized by five coefficients ($a$ ,$b$ ,$c$ ,$c_1$ ,$c_2$), which can be learned by the backpropagation algorithm. These coefficients are the only additional parameters that we learn for each neuron. $a$ ,$b$ and $c$ are the scalars that we use to parametrize the ODE, and $c_1$ and $c_2$ represent the initial conditions of the ODE's solution. $g(t)$ is a regulatory function that we have called the "core activation function". Mainly to simplify the math, and because it is a standard practice in control theory, we have set $g(t)$ to the Heaviside step function. Therefore, the ODE that we have chosen is the following:

$$ay''(t) + by'(t) + cy(t) = u(t), \quad \text{where } u(t) = \begin{cases} 0 & x \leq 0 \\ 1 & x > 0 \end{cases} \tag{1}$$

In engineering and physics, such a model is often used to denote the exchange of energy between mass and stiffness elements in a mechanical system, or between capacitors and inductors in an electrical system (Ogata & Yang, 2002). Interestingly by using the solutions of this formulation as activation functions, we can gain a few key properties: approximation or reduction to some of the standard activation functions such as sigmoid or ReLU; the ability to capture oscillatory forms; and, exponential decay or growth. The latter commonly appear in health-care problems.

### 3.2 THE LEARNING ALGORITHM

For fixed $a$, $b$ and $c$, the solution of the differential equation will be $y = f(t; a, b, c) + c_1 f_1(t; a, b, c) + c_2 f_2(t; a, b, c)$ for some functions $f$, $f_1$, $f_2$. $y$ lies on an affine space parametrized by scalars $c_1$ and $c_2$ that represent the initial conditions of the solution. As described in the following two subsections, our learning algorithm has two main parts: solving the differential equations once, and using a backpropagation-based algorithm for jointly learning the network weights and the five parameters of each neuron.

#### 3.2.1 CLOSED-FORM SOLUTIONS

First, we solve the differential equations parametrically and take the derivatives of the closed-form solutions: $\frac{\partial y}{\partial t}$ with respect to its input $t$, and $\frac{\partial y}{\partial a}$, $\frac{\partial y}{\partial b}$, $\frac{\partial y}{\partial c}$ with respect to parameter $a$, $b$, $c$. Moreover,

---

[1]Up to computational precision limitations.

the derivative with respect to $c_1$ and $c_2$ will be $f_1$ and $f_2$, respectively. This is done once. We solved the equations and took their derivatives using the software package Maple (Maple 2018). Maple also generates optimized code for the solutions, by breaking down equations in order to reuse computations. [2]. Although we used Maple here, this task could have been done simply by pen and paper (although more time consuming).

**Singularity of solutions.** If one or two of the coefficients $a$, $b$ or $c$ are zero, then solution of the differential equation falls into a singularity subspace that is different than the affine function space of neighboring positive or negative values for those coefficients. For example, for $b = 0$ and $a * c > 0$, the solution will be $y(t) = \sin\left(\frac{\sqrt{c}t}{\sqrt{a}}\right) c_2 + \cos\left(\frac{\sqrt{c}t}{\sqrt{a}}\right) c_1 - \frac{u(t)}{c}\left(\cos\left(\frac{\sqrt{c}t}{\sqrt{a}}\right) - 1\right)$, but for $b = c = 0$, we will have $y(t) = 1/2 \frac{u(t)t^2}{a} + c_1 t + c_2$. We observe that changing $c > 0$ to $c = 0$ will change the resulting activation function from a pure ocsillatory form to a (parametric) leaky rectified quadratic activation function. Our learning algorithm allows the activation functions to jump over the singularity subspaces. However, if they fall into a singular subspace, the derivative with respect the parameter that has become zero, and will be zero for the rest of training. Therefore, the training algorithm will continue to search for a better activation function only withing the singular subspace.

In practice, for some hyperparameter $\epsilon$, if any one of $a$, $b$ or $c$ is less than $\epsilon$, we project that value to exactly zero, and use the corresponding solution form the singular sub-space. We do not allow $a = b = c = 0$, in this rare case we force $c = \epsilon$. During the learning process at most two of $a$, $b$ and $c$ can be zero, which creates seven possible subspaces (with $a, b, c \in \{\mathbb{R} - \{0\}, \{0\}\}$) that are individually solved. Similarly, when $b^2 - 4ac$ is close to zero, the generic solution will be exponentially large, therefore if $-\epsilon < b^2 - 4ac < \epsilon$, we explicitly set $b = \sqrt{(4ac)}$ to stablize the solution and to avoid large function values.

**Approximation of Dirac's delta function.** The derivative of activation function with respect to $t$ will have Dirac's delta function $\delta(t)$ which is the derivative of the Heaviside function. In the parametric derivatives, we substituded the delta function with its approximation $s * e^{-s*t}/(1 + e^{-s*t}))^2$, which is the derivative of $\sigma(s * t) = 1/(1 + e^{s*t}))$. This approximation is a commonly used in practice for the delta function (Zahedi & Tornberg, 2010). The larger $s$ is, the more accurate the approximation of delta function will be.

In all of our experiments, we set $\epsilon = .01$, and $s = 100$, although tuning might improve the results.

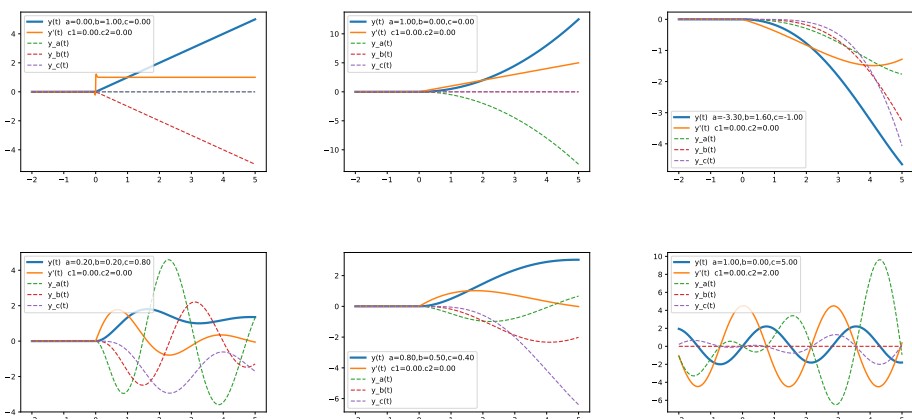

Figure 1: A sample set of DifEN activation functions and their derivatives. The bold blue line is the activation fucntion, and the orange solid line is its derivative with respect to $t$. The dashed lines are its derivatives with respect to $a$, $b$ and $c$. First and second on the top row from left are ReLU and ReQU. The bump in the derivative of ReLU is an artifact of approximating Dirac's delta.

---

[2]The code will be accessible, upon request.

### 3.2.2  GRADIENT DESCENT-BASED LEARNING

We adopt regular backpropagation to update the values of parameters $a$, $b$, $c$, $c_1$ and $c_2$ for each neuron, along with using $\frac{\partial y}{\partial t}$ for updating network parameters $w$ (input weights to the neuron), and for backpropagating the error to lower layers.

We initialize network parameters using current best practices with respect to the layer type (e.g. linear layer, convolutional layer, etc.). We initialize parameters $a$, $b$, $c$ for all neurons with a random positive number less than one, and strictly greater than zero. We initialize $c_1 = c_2 = 0.0$. To learn the parameters $\theta = [a, b, c, c_1, c_2]^T$ along with the weights $\mathbf{w}$ on input values to each neuron, we deploy a gradient descent algorithm. Both the weights $\mathbf{w}$, as well as $\theta$ are learned using the conventional backpropagation algorithm with Adam updates (Kingma & Ba, 2014).

During training, we treat $a, b, c, c_1$ and $c_2$ like biases to the neuron (i.e., with input weight of $1.0$) and update their values based on the direction of the corresponding gradients in each mini-batch.

### 3.3  DIFEN IS UNIVERSAL APPROXIMATOR

Feedforward neural networks with monotonically-increasing activation functions are universal approximators (Hornik et al., 1989; Barron, 1993). Networks with radial basis activation functions that are bounded, increasing and then decreasing are also shown to be universal approximators (Park & Sandberg, 1991). In this subsection we show that DifEN is also a universal approximator.

**Lemma 1.** *If $b^2 - 4ac < 0$, the solutions of $ay''(t) + by'(t) + cy(t) = u(t)$ will oscillate with frequency $\omega = \frac{\sqrt{4ac - b^2}}{2a}$, and in particular, if $b = 0$, then $\omega = \sqrt{\frac{c}{a}}$.*

*Proof.* If $b^2 - 4ac < 0$, the roots of the characteristic equation of the ODE will be $\frac{-b \pm i\sqrt{4ac - b^2}}{2a}$, where $i = \sqrt{-1}$. By substituting the solutions in Euler's formula the resulting sine and cosine functions will have frequency $\omega = \frac{\sqrt{4ac - b^2}}{2a}$. In particular, if $b = 0$, we'll have $\omega = \sqrt{\frac{c}{a}}$.  □

Moreover, the solution of the ODE $ay''(t) + cy(t) = u(t)$ for $ac > 0$ is $y(t) = \sin\left(\frac{\sqrt{c}t}{\sqrt{a}}\right) c_2 + \cos\left(\frac{\sqrt{c}t}{\sqrt{a}}\right) c_1 - \frac{u(t)}{c}\left(\cos\left(\frac{\sqrt{c}t}{\sqrt{a}}\right) - 1\right)$. We will use this in the proof of the following theorem:

**Theorem 2.** *Differential equation network is a universal approximator.*

*Proof.* By Lemma 1, for $b = 0$ and $c = 1$, the oscilation frequency of the activation function of a neuron will be $\omega = 1/\sqrt{a}$. Therefore, by varying $a \in \mathbb{R}^+$, a neuron activation function can generate all oscillation frequencies. Now, consider a neural network with one hidden layer of differential equation neurons and one linear output layer. For a target function $h(t)$, we take the fourier transformation of $h(t)$, and find the set of present frequencies $\Omega$. For each present frequency $\omega_i \in \Omega$, we add two differential equation neuron $i_1$ and $i_2$ to the hidden layer with $c = 1$, $b = 0$ and $a = 1/\omega_i^2$. Let $\beta_i$ and $\gamma_i$ be the corresponding coefficient of the sine and cosine functions from the Fourier transformation. Then, we let $c_{1i_1} = \gamma_i, c_{2i_1} = \beta_i$ and $c_{1i_2} = c_{2i_2} = 0$, and corresponding weights $w_{i_1}$ and $w_{i_2}$ from neurons $i_1$ and $i_2$ to the output linear layer to be equal to $1.0$ and $-1.0$, respectively. This way, the two neurons will cancel the particular solution term of the function, and we'll have: $y_{i_1}(t) - y_{i_2}(t) = \beta_i \sin(\omega_i t) + \gamma_i \cos(\omega t)$. By construction, after adding all frequencies from the Fourier transformation of $h(t)$, the neural network will output $\sum_{i \in \Omega} \beta_i \sin(\omega_i t) + \gamma_i \cos(\omega t) = h(t)$.  □

There are multiple other ways to prove this theorem. For example, since the sigmoid, ReLU, and ReLU-like functions are among the solutions of differential equations, we can directly apply results of the Stone-Weierstrass approximation theorem (De Branges, 1959).

### 3.4  A GEOMETRIC INTERPRETATION

The solution set of a differential equation forms a functional manifold that is affine with respect to $c_1$ and $c_2$, but is nonlinear in $a$, $b$, and $c$. Clearly, this manifold has a trivially low dimensional representation in $\mathbb{R}^5$ (i.e., $\{a, b, c, c_1, c_2\}$). Gradient descent changes the functionals in this low

dimensional space, and the corresponding functional on the solution manifold is used as the learned activation function. Figure 2 attempts to visually explain how, for example, a ReLU activation function transforms to a cosine activation function.

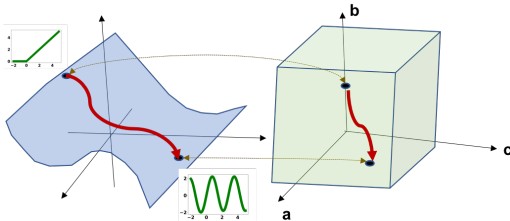

Figure 2: Left: The solutions of $ay'' + by' + cy = u(t)$ lie on a manifold of functions. Right: Every point on this manifold can be equivalently represented as a point on a 5-dimentional space of $\{a, b, c, c_1, c_2\}$ (three shown in the cartoon). The red arrow shows a path an initialized function takes to be gradually transformed to a different one.

One DifEN neuron approximating the sine functions in Figure 3 shows an empirical example of this scenario where we initialized the activation functions with random parameters, and the learned differential equation coefficients $\{a, b, c\}$ and initial condition coefficients $\{c_1, c2\}$ perfectly represented a sine function after training. In contrast, the learned model by ordinary fixed activation FFNNs were much less accurate with significantly larger networks. Figure 5. in the appendix material visually shows how changing a parameter changes the activation function's behavior.

### 3.5 REDUCTION TO COMMON ACTIVATION FUNCTIONS

Two common activation functions are the sigmoid $\sigma(t) = \frac{1}{1+e^{-t}}$, and the rectified linear unit $\text{ReLU}(t) = \max(0, t)$. The sigmoid function is a smooth approximation of the Heavyside step function, and ReLU can be approximated by integrating sigmoid of $s * t$ for a large enough $s$: $\max(0, t) \approx \int_{-\infty}^{t} \frac{1}{1+e^{-sz}} dz = \log(1 + e^{st})/s$. Equivalently, $y(t) = \log(1 + e^t) + c_1 \approx \text{ReLU}(t) + c_1$ will be a solution of the following first order linear differential equation: $y'(t) = \frac{1}{1+e^{-st}} \approx u(t)$

We can set $g(t)$ to $\sigma(t) = \frac{1}{1+e^{-st}}$, or to the step function $u(t)$. For $g(t) = \sigma(t)$, the particular solutions of this differential equation when $a \neq 0$ involve the Gauss hypergeometric and $Li_2$ functions, which are expensive to evaluate. Fortunately, if we set the right hand side to $u(t)$, then the particular solutions will only involve common functions such as linear, logarithmic, exponential, quadratic, trigonometric, and hyperbolic functions.

In practice, if the learning algorithm decides that $a$ and $b$ should be zero, we use $g(t) = \frac{1}{1+e^{-st}}$ (i.e. $y(t) = \frac{1}{c*(1+e^{-st})}$). Otherwise, we use a step function to avoid complex-valued solutions that involve special mathematical functions, and particularly for the reasons mentioned in the previous subsection. With these conditions in place, if $a = 0$, $b = 0$, and $c = 1$, we recover the sigmoid function; if $a = 0$, $b = 1$, and $c = 0$ we recover the ReLU function; if $a = 1$, $b = 0$, and $c = 0$ we obtain a parametric leaky rectified quadratic form $y = \text{ReLU}(t)^2 + c_1 t + c_2$ (similar to parametric leaky ReLU (He et al., 2015; Xu et al., 2015)), which is the solution of $y''(t) = u(t)$. When $b^2 - 4ac < 0$ we observe oscillatory behaviour. Depending on the sign of $b$, this can be decaying or exploding, but when $b = 0$ observe a purely oscilatory behavior.

The above-mentioned cases are only a few examples of solutions that could be chosen. The point to emphasize is that an extensive range of functions can be generated by simply varying these few parameters. (Figure 1 illistraits several examples.)

## 4 RESULTS AND DISCUSSION

We have conducted several experiments to evaluate the performance and compactness of differential equation networks.

### 4.1 TOY REGRESSION DATASETS

We ran two tests on toy regression problems. In these tests we compared fixed activation networks using ReLU, LeakyReLU and SELU activations to a significantly smaller DifEN.

**Sine.** We trained on two periods of a sine function and extrapolated half of one period of unseen values. As seen in Figure 3, a single differential equation neuron can learn the function almost perfectly, while the fixed activation baseline networks fail to fit comparable to the training data, even with a 10x larger network.

**A more sophisticated function.** Next, we fit a more challnging function ($y = (\sin(t) - \cos(2t)^2)/2 + 4*(1 + \arccos(\sin(t/2)))/3$). A DifEN with 25 neurons in one hidden layer fits the function much more accurately than the fixed activation baseline networks with 250 neurons as shown in Figure 4.

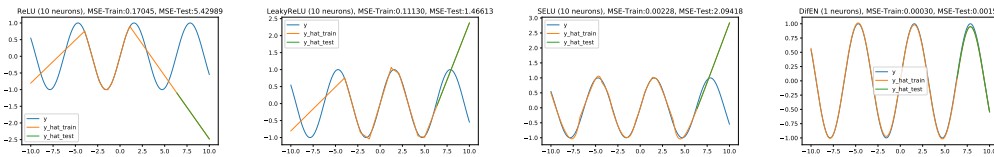

Figure 3: Learning the sine function. The differential equation neuron learns the function significant;y better than larger baselines.

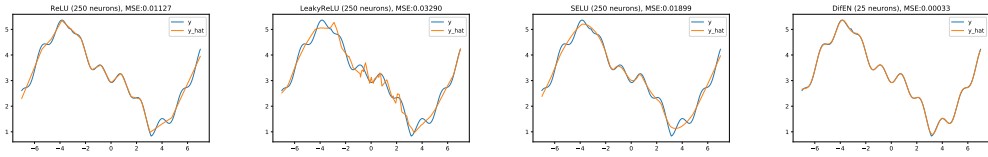

Figure 4: Fitting an arbitrary mathematical function.

### 4.2 CLASSIFICATION

We compared the performance of DifEN with baselines on MNIST handwritten digit dataset. We compared the performance of fixed activation convolutional neural networks equipped with ReLU, SELU, or Swish (Ramachandran et al., 2017) activation functions to a smaller DifEN. The fixed activation networks all used the following architecture: Three convolutional layers consisting of 20 5x5 filters, 40 5x5 filters, and 60 4x4 filters respectively. A pooling layer and dropout layer was applied after the second and third activation layers, and two fully connected layers followed with 200 neurons in the first fully connected layer. The DifEN network comprised two convolutional layers with 20 5x5 filters and 40 3x3 filters, respectively. One pooling and one dropout layer were applied after the second activation. One fully connected layer was used at the end as opposed to two for the fixed activation networks, reducing the number of parameters further. Dropout probability, batch size, epochs and learning rate were consistent across all networks. The goal here was to test the assumption that a DifEN should be able to at least match the performance of a significantly larger, fixed activation network, thereby achieving network compression without sacrificing performance. For these experiments, we ran three-fold cross-validation and reported the mean result for each network.

While the performance boost here is negligable, the notable achievement is that the CNN with DifEN activations was able to learn a representation on par with that of the significantly larger fixed activation networks.

### 4.3 REGRESSION

We compare the performance of DifEN to that of fixed activation DNNs applied to a standard diabetes regression dataset. In this section we compared the performance of a simple, one hidden layer FFNN

Table 1: Comparison of performance on MNIST by activation. Compactness DifEN performance is on par with significantly larger fixed activation networks

| Activation | ReLU | LReLU | SeLU | Swish | DifEN |
|---|---|---|---|---|---|
| Accuracy | 0.9891 | 0.9890 | 0.9903 | 0.9918 | **0.9919** |

equipped with fixed activations to that with DifEN neurons. Again, we use 3-fold cross validation and report the average performance. The following experiments were conducted using the open source diabetes regression data.

Table 2: Comparison of performance on diabetes regression dataset by network size and activation

| size | DifEN | ReLU | LReLU | SeLU | Swish |
|---|---|---|---|---|---|
| 1 | **2490.781** | 7391.783 | 6977.1 | 6289.9 | 4298.493 |
| 2 | **2446.003** | 3759.527 | 4562.308 | 3793.336 | 3249.608 |
| 4 | **2412.504** | 2931.891 | 2720.555 | 2912.025 | 2839.323 |
| 8 | **2313.98** | 2465.16 | 2398.2 | 2488.361 | 2664.854 |
| 16 | **2117.47** | 2334.454 | 2357.557 | 2165.465 | 2236.137 |

We ran the test in Figure 3, numerous times with different initializations, each time the differential transformed itself to have a sine-like solution. This supports that DifEN activation functions transform during the training process. We also demonstrated the capability of DifENs to learning complex concepts, and with a significantly reduced network size. Table 2 shows a DifEN can perform on par, or better than a network with over 2x the number of parameters when compared to a fixed activation network. Moreover, DifENs can learn a better approximation when compared to a network with fixed activations throughout. The ability of DifENs to achieve top-notch performance with a compact representation makes them a good candidate for on-device applications. Moreover, DifENs seem well suited to applications that require the capabilities of a big neural network, but are currently limited by memory, or where latency issues are a factor, such as space applications and robotics.

## 5  CONCLUSION

To the best of the authors' knowledge, the machine learning community has yet to explore differential equations in neural networks, dictionary learning (Mairal et al., 2009a;b; Zhang & Li, 2010), or kernel methods (Shawe-Taylor & Cristianini, 2004; Schölkopf et al., 1999). While the presented model, algorithms and results in this paper are the first application of ODEs in neural networks, they show a promising and successful example of the potential of differential equations in the development of new machine learning algorithm.

In this paper we introduced Differential Equation Networks (DifEN). We have showcased the ability of DifENs to learn complicated concepts with a compact network representation. We have demonstrated DifENs' potential to outperform conventional DNNs across a number of tasks, and with a reduced network size. Modern DNNs achieve performance gains in large by increasing the size of the network, which is not a sustainable trend. DifENs represent a new spin on deep learning, and possibly a way to expand the capabilities of machine learning and on-device AI. The proposed algorithm introduces a mechanism for learning the activation functions for each neuron in a network, and empirical results support a promising advancement in learning complex concepts in a compact representation.

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

## A The Euler-Lagrange Equation in Calculus of Variations

As practiced in calculus of variations, a functional is a function of functions, and it is usually expressed as a definite integral over a mapping $L$ from the function of interest $y$ and its derivatives at point $t$ to a real number: $\mathbf{J}[y] = \int_{t \in \mathcal{T}} L(t, y, y', y'', \ldots, y^{(n)})dt$. $y(t)$ is the function of interest, and $y', y'', \ldots, y^{(n)}$ are its derivatives up to the $n$th order.

Intuitively, $L$ is a combining function, which reflects the cost (or the reward) of selecting the function $y$ at point $t$, and the integral sums up the overall costs for all $t \in \mathcal{T}$. Therefore, the functional $\mathbf{J} : (\mathbb{R} \to \mathbb{R}) \to \mathbb{R}$ can represent the cost of choosing a specific input function. The minimum of the functional is a function $y^*(t) = \arg\min_y \mathbf{J}(y)$ that incurs the least cost. Among other methods, $y^*(t)$ can be found by solving a corresponding differential equation obtained by the Euler-Lagrange equation (Gelfand & Fomin, 1963; Gelfand et al., 2000). In particular, the extrema to $\mathbf{J}[y] = \int_{t_1}^{t_2} L(t, y, y', y'')dt$ are the same as the solutions of the following differential equation:

$$\frac{\partial L}{\partial y} - \frac{d}{dt}\frac{\partial L}{\partial y'} + \frac{d^2}{dt^2}\frac{\partial L}{\partial y''} = 0 \tag{2}$$

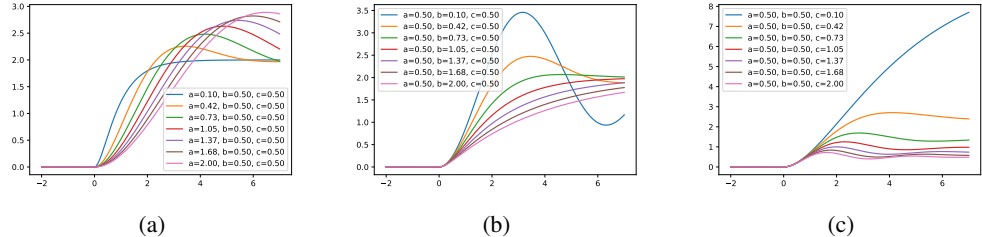

(a)                          (b)                          (c)

Figure 5: The spectra of functions generated by varying one of $a, b, c$, and fixing the other two with $c_1 = c_2 = 0$.

## B    EXAMPLE SPECTRA OF POSSIBLE ACTIVATION FUNCTIONS

Figure 5 shows how changing a coefficient in the low dimensional differential equation space representation will affect the resulting functional on the manifold.

## C    COMPARATIVE CONVERGENCE OF RESULTS FOR THE DIABETES DATASET

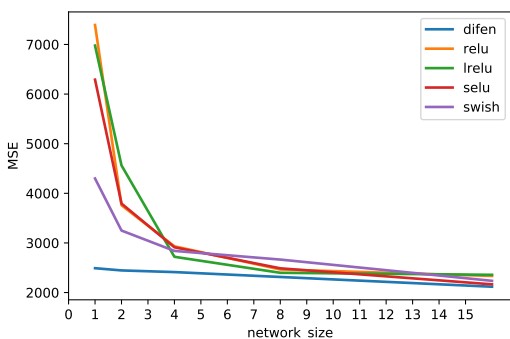

Figure 6: Diabetes Regression Model Convergence Comparison

Figure 6 above shows that the differential equation network achieves performance on par with significantly larger fixed activation networks. We see that the fixed activation networks do not surpass the single neuron DifEN performance untill they 8 or more neurons.

