# OpenReview forum: "Differential Equation Networks"
_ICLR.cc/2019/Conference_

### Official Review · AnonReviewer3 · 2018-11-02
**Differential Equation Networks**

**Rating:** 5
**Confidence:** 4

**Review:**

This paper proposes an intriguing idea, that of using solutions to a differential equation as activation functions in a neural network. Coefficients of the differential equation (five parameter equations in implementations done) are trainable, realising different activation functions on different nodes. Back propagation is used with the ADMA optimiser to train the parameters of the DE as well as the remaining weights. What the network implements is shown to achieve universal approximation by considering a second order differential equation producing different sinusoidal functions which can add up to a desired function. The paper is written clearly and easy to follow. The idea is novel. The work is illustrated on three problems: (a) a toy dataset, (b) MNIST classification problem and (c) a regression task from a diabetes problem.
While the idea is novel and the paper is clear, the empirical work presented in the paper does not go far enough to be supportive of its acceptance. Firstly, Tables 1 and 2 do not provide any uncertainty in results. Simply saying accuracy of one method is marginally higher (in the second decimal place in Table 1) than another method is not persuasive. This is particularly so when no training set results are reported. I would strongly urge to report uncertainty coming from cross validation (three fold is too small; the data is large enough to do ten-fold). Second, some sort of error analysis has to be carried out to understand how the improved performance is attributable to the new idea being advanced. Confusion matrices on the classification problem might help. Is there a specific part of the task in which the new method separates characters that the more classic ones fail to do? Similar criticisms apply to the regression task; is the improvement across all examples or localized to some particularly hard ones; this is an issue when comparing (squared) errors because a few outliers can dominate the evaluation/comparison.
In summary, the paper has a novel idea, but has to be better developed in its empirical part.

---

### Official Review · AnonReviewer2 · 2018-11-02
**Interesting concept, needs stronger backing empirically**

**Rating:** 4
**Confidence:** 4

**Review:**

This paper proposes the use of structured activations based on ordinary differential equations, as an activation in neural network architectures.
There are validations that the approach discovers different activations, and comparisons to a variety of other architectures with fixed activations. In general, I would like to see additional contextualization of your work with other approaches which learn activation functions. How does your approach differ from max-out and Agostinelli? References to Network in Network, Lin et al and even Hypernetworks, Ha et al would also be helpful. Some of these papers are cited, but only the comparison to Ramachadran et. al. is directly discussed and the methodological difference there is not true for the case of Maxout.

The paper indicates that the ODE should be solved before beginning to use the network. Should this process be performed once per network design? Once per dataset? What happens if instead of solving f1 and f2, these are set randomly? How is this actually solved in the case of the MNIST convnet?

The experiments were useful in demonstrating the proposed method. However, some discussion and comparison to other learned activation functions would be helpful (for instance, one could perform similar experiments as in Maxout). Performance on larger datasets, as seen in Swish, would make the results more compelling.

The MNIST experiments shown are also pretty far from standard baselines. See for example the benchmark performance in Maxout, which also references an architecture from Jarrett et. al., 2009 which is quite similar to the baseline architecture, but ~.5% error. It isn't necessary to get the absolute best performance with a new activation, just show that the proposed doesn't actively hurt and enables new interpretation or direction. But as it stands, it isn't possible to tell from the experiments if the proposed method has serious limitations, because the baselines on MNIST are below where they should be.

In general, a larger test suite that compares on more standard datasets is necessary to really prove out this idea, or see if there are problems with cases on larger datasets. CIFAR10 at a minimum would be a key addition as well as other datasets (besides the diabetes dataset shown) where there can be direct comparison to existing work. Currently, only MNIST is filling that role. Many of the cited / compared work (PReLU, Swish, LReLU, SELU) has a broad suite of benchmarks, all on datasets with existing numbers tuned by the authors of respective past papers.

What are the downsides of this method? Tradeoffs in memory and training time should be discussed in detail if application to low power hardware is a real application area, perhaps along with an inclusion of the time / effort required for solving the ODEs in Maple. A difference of 2x+ in parameter count may not be a difference if the computational time is much worse. Can you consider a case where "normal" architectures don't fit in memory, but one based on ODE activations will? The paper directly discusses mobile and low footprint deployments, but without discussion of the computational overhead and complexity it is speculative, especially when there are also numerous methods for compressing or distilling very large architectures to much smaller sizes, as well as small models which directly achieve high performance. A few relevant methods are linked below.

MobileNets - https://arxiv.org/abs/1704.04861
ENet - https://arxiv.org/abs/1606.02147

In the Network Compression section, the paper fails to discuss a number of successful foundational and modern network compression techniques that would improve the argument, including:
Optimal brain damage  - “removing unimportant weights can actually improve performance” - https://papers.nips.cc/paper/250-optimal-brain-damage
Deep compression: Compressing deep neural network with pruning, trained quantization and huffman coding -  pruned state-of-the-art CNN models with no loss of accuracy - https://arxiv.org/abs/1510.00149
Bayesian Compression for Deep Learning - https://arxiv.org/abs/1705.08665
Practical Variational Inference for Neural Networks - http://papers.nips.cc/paper/4329-practical-variational-inference-for-neural-networks

There are repeated claims of first use of ODE in neural networks, which is frankly false. Though the specific use proposed here may be new, neural networks and ODEs have been used together many times. Clarifying what particular usage of ODE inside this setting is novel would be much better than a broad claim such as "While the presented model, algorithms and results in this paper are the first application of ODEs in neural networks...". Much of this work has been about controlling or solving ODEs, but particularly the setting of Meade Jr. et. al. strongly resembles a "neuron" in this architecture, so a discussion of the relevant differences would be useful. In addition Neural Ordinary Differential Equations allows the end-to-end training of ODEs in larger models, which also closely resembles the use of ODEs here.

Artificial Neural Networks for Solving Ordinary and Partial Differential Equations - https://ieeexplore.ieee.org/document/712178/
Solution of nonlinear ordinary differential equations by feedforward neural networks - https://www.sciencedirect.com/science/article/pii/089571779400160X
Neural Ordinary Differential Equations - https://arxiv.org/abs/1806.07366

Overall, a stronger focus on empirical results on comparable datasets would be beneficial, especially larger tasks. If larger tasks are not possible, a description of what it may take to "scale up" would be useful. The written focus on novelty detracts from the presentation, and a discussion of neural ODE methods (whether acting as activations, or solvers) would serve as good background material. If compute / performance in low footprints or mobile hardware is a focus, it should be described and tested. If lower parameter count is a perceived benefit, a more direct exploration and discussion of parameter count settings for this architecture and baselines would also be useful. Particularly, hyperparameters become very important in small architectures, so "Dropout probability,
batch size, epochs and learning rate were consistent across all networks" is not a positive (presuming the authors have likely tuned toward their own architecture). Baselines should be given equal treatment and tuning in order to compare "best-on-best" performance.

The description of universal approximation, visualization of the adaptivity of the method, and background are all very nice. My concerns come primarily to relation to prior and relevant work, strength of relevant experimentation, and claims of application and novelty / "first past the post".

——

Minor Nitpicks:
Page 1: In the sentence - “researchers have introduced highly effective network structures such as convolutional neural networks”, it seems inconsistent to cite a foundational paper for CNNs and not RNNs.

Page 2: It seems like there is a word missing here - “The size of a neural network is delineated its number of hidden neurons and their interconnections, which together determine the network’s complexity”

There seems to be a missing word in “3.3 DIFEN IS UNIVERSAL APPROXIMATOR” .

Numerous spelling errors should be corrected -
3.1 differentiatial
4.1 challnging
Figure 1 - fucntion

---

### Official Review · AnonReviewer1 · 2018-11-03
**Interesting approach but limited experiments and insights**

**Rating:** 5
**Confidence:** 3

**Review:**

Personal Expertise:
The reviewer has extensive practical and theoretical experience with deep networks, different activation functions, as well as some practical experience using deep networks to model physical phenomena which are different inverse problems than the more common perception modeling. However, the reviewer is not knowledgeable in using ODEs for deep networks.
Contributions of the paper :
The contribution of the paper is in proposing a learnable activation function in form of an ODE which can help to better model highly oscillatory and irregular functions more efficiently. This can be potentially useful for special applications in inverse problems where (by field knowledge) we know highly non-linear and specific activation functions can be more reasonable than the common ReLU and its recent variants.

Quality and composition:
The composition of the theoretical part of the paper is clear. The experimental part is very limited though and does not include all the necessary details. Many of the details in the initial sections can be taken to the appendices to make room for more empirical studies.

Novelty, related works:
The work seems novel in proposing the specific activation function but in general there are many other works that propose learnable or more elaborate activation functions, neurons, or local parts of networks. It is not so clear how this work is different from those and nor is compared to those works. This includes networks in networks, maxout networks, capsule networks, etc.

Critique of the theories and experiments:

Theoretical Design:
The theoretical design and derivation of the paper seem correct, although the reviewer is not an expert on this topic. However, it does not clearly mention why the ODE is not designed and solved for each problem separately. Should there be different design choices for y for each task/dataset? Why not solving the ODE during the training as well? If we are solving the ODE only once and based on some initialization of coefficients, it seems to be equivalent to designing a learnable activation function such as leaky ReLU. In that regard, one could call leaky-ReLU a DifEN?

Experimental Setup:
The motivation of the new activation function is for specific use-cases where oscillatory or decaying functions are to be modelled. In that respect, the experimental setup is quite limited and inconclusive.
- MNIST experiment: to conclusively evaluate the performance of the proposed activation function, it is important to try fixed activation functions on the same architecture as the DifEN and vice versa.
- Diabetes regression experiment: since the task is not a well-studied regression task, more experiments on various datasets are required to make a conclusion.
- The learnable activation function can potentially make the network more prone to overfitting, this needs to be tested thoroughly.
- An important application of the proposed activation function is mentioned to be for model compression. That should be properly tested on problems with large sets of parameters (such as ImageNet networks) and observe if the performance drop due to a decrease in the number of parameters in a standard network is sharper than that of a network with DifEN activations.
- More tasks and more analysis should be performed for the real-world tasks that the authors mention as the motivation of this work (specifically medical diagnosis or predictions). The analysis should demonstrate and give insight on the extra generalization power that the new activation function brings to those problems.
- Following on the previous point, it is important to empirically demonstrate for which applications DifEN is useful.
- Some analyses are missing on when a neuron can accentuate the problem of vanishing and/or exploding gradients in certain configurations of the DifEN parameters. It seems like the situation can arise during the training where the activation function become too steep or too saturated.

Summary judgment:
All in all, I think the proposed work has some potential in specific applications, however, the experimental setup does not give a clear and conclusive message of where and how the new activation functions are useful.

---

### Meta-Review · Area_Chair1 · 2018-12-14
**Rejection, reviewer concerns not addressed**

**Confidence:** 5
**Recommendation:** Reject

**Metareview:**

The reviewers unanimously agreed the paper did not meet the bar of acceptance for ICLR. They raised questions around the technical correctness of the paper, as well as the experimental setup. The authors did not address any reviewer concerns, or provide any response. Therefore, I recommend rejection.